# Changes in Alcohol Consumption after 1 Year of the COVID-19 Pandemic: A Cross-Sectional Study in a Region of France

**DOI:** 10.3390/ijerph192215049

**Published:** 2022-11-15

**Authors:** Pierre-Antoine Villette, Olga Lyonnard, Camille Trehu, Marie Barais, Delphine Le Goff, Bernard Le Floch, Antoine Dany, Morgane Guillou Landreat

**Affiliations:** 1Medicine Faculty, University of Bretagne Occidentale, 29200 Brest, France; 2ER 7479 SPURBO, Department of General Practice, University of Western Brittany, 29200 Brest, France; 3Addictology Liaison Department, University Hospital of Brest, Bd Tanguy Prigent, 29200 Brest, France; 4HUGOPSY Network, 29200 Brest, France

**Keywords:** COVID-19, alcohol consumption trends, France

## Abstract

Background: There is conflicting evidence on how the COVID-19 pandemic changed patterns of alcohol consumption. While some studies have suggested that alcohol consumption decreased at the beginning of the pandemic, there are limited data for a longer period. The objective of this study was to investigate changes in alcohol consumption 1 year after the onset of the COVID-19 pandemic in France, and to identify vulnerable subgroups in a French adult population. Methods: This was a single-center, cross-sectional, descriptive study. Self-reported changes in alcohol consumption were collected from 2491 respondents in a survey carried out in western Brittany from 18 January to 9 March 2021. Results: Of respondents, 27.64% reported that they had increased their alcohol consumption, 14.7% had decreased, 3.94% had ceased, and 53.72% reported no change in their alcohol consumption. Increased alcohol use was associated with male gender, age 26 to 44 years, living with a family, not being a health professional, having had a physical or psychological health problem during lockdowns, smoking tobacco, and using cannabis. Reduced alcohol use or cessation was associated with male gender, age 18 to 25 years, living in Brest, living alone, and using cannabis. Conclusions: Our study suggests that during the COVID-19 pandemic, a significant number of people increased their alcohol consumption in France, even outside lockdowns. These results should encourage health professionals and public authorities to implement more specific prevention measures to limit the risks associated with alcohol consumption.

## 1. Introduction

Alcohol consumption is responsible for 3 million deaths per year worldwide. It is the prime preventable risk factor for death and disability among 15–49-year-olds [1].

According to the WHO, Europe is the region with the highest alcohol consumption [2]. In France, the level of alcohol consumption remains high [3]: according to the French National Public Health Agency, 23.6% of people aged 18 to 75 exceed the accepted consumption guidelines, and alcohol was responsible for 41,000 deaths in France in 2017 [4]. Alcohol consumption could evolve according to environmental factors, for instance, with the recent environmental changes resulting from the pandemic. The SARS-CoV-2 epidemic appeared in late 2019 in China in Wuhan before spreading worldwide. On 11 March 2020, the WHO reclassified the epidemic as a pandemic, then termed COVID-19 [5]. In order to stop the exponential spread of the virus and to preserve the healthcare system, the French government implemented unprecedented measures to limit social interactions among individuals, which disrupted the daily life of the French population. Two main lockdown periods were decided: the first in spring 2020 (March to May), during which all “non-essential” businesses, bars, and restaurants were closed, as were schools, and the population was required to stay at home with restrictions on movement; during the second lockdown from 30 October 2020 to 15 December 2020, schools remained open, and it was possible for some people to go to work, but the other measures were similar.

These two successive waves raised fears of a third wave in the form of an epidemic of psychological disorders in the population [6]. Restaurants and social venues where alcohol consumption was possible were closed, and social gatherings were severely restricted during these periods, but alcohol was still widely available in supermarkets, so home consumption became a concern. Among the main motivations for alcohol consumption, the need to counteract depressive affects or stress is prominent, and many studies have shown that the co-occurrence of these disorders potentiates the negative effects in terms of prognosis [7,8,9,10]. In a vicious circle, it is feared that the climate of anxiety could mechanically increase the factors favoring alcohol consumption.

Existing studies exploring the initial impact of the COVID-19 pandemic on alcohol consumption in the general population present conflicting results. Some studies suggest that more people increased their alcohol consumption than reduced it [11,12]. This increase could be seen as a coping strategy to manage psychological distress. However, other studies, in contrast, indicate that alcohol consumption may have decreased at the population level during the first months of the COVID-19 pandemic [13,14]. According to these results, the restriction of access to the usual places of alcohol consumption, as well as the reduced affordability as a result of increased unemployment and financial insecurity, could have led to a reduction in the levels of alcohol consumption in the general population [15,16]. A recent meta-analysis published by Kilian et al. [17] suggested that during the first months of the COVID-19 pandemic, the number of people who reduced their alcohol consumption in Europe was greater than the number of people who increased their consumption. Concerning the evolution in France, Guignard et al. reported that during the first lockdown, 10.7% of the respondents reported having increased their alcohol consumption and 24.4% reported having decreased it [18].

To our knowledge, few longer-term studies have been conducted in France to explore the evolution of alcohol consumption levels during the pandemic. Consequently, the objective of this study was to evaluate the evolution of alcohol consumption 1 year after the onset of the COVID-19 pandemic in a region of France (Brittany), and to determine the factors associated with an increase or decrease in alcohol consumption.

## 2. Materials and Methods

### 2.1. Study Design/Sample

A cross-sectional study was conducted in the general population. This study used an online self-administered questionnaire. This questionnaire was developed by the study’s scientific committee, which included doctors specializing in addictive disorders, general practitioners, and methodologists. The questionnaire was first tested on a small sample of people, to make sure that the questions were well understood, and that the process and time needed to complete it were acceptable.

Circulation of the questionnaire 

The inclusion criteria were: being aged over 18 years, command of the French language (oral and written), and consent to participate. As the intention was to target the adult general population, only under-age participants were excluded.

The questionnaire was widely disseminated to the general population through several media, either email lists with a link to the study, or using a QR code which was linked to the questionnaire. We obtained agreement from the Brest university hospital authorities, the city of Brest and the university to disseminate the questionnaire to employees via their mailing lists. The study was disseminated through the press (communication in the local press and on local television) and on the Internet (social networks and mailing lists of the investigators, and also the city of Brest, the University Hospital of Brest, the University of Western Brittany, and in particular the Citizen’s University of Health Prevention of Western Brittany, the North Finistere Addictology Network, and the Department of General Medicine). It was also disseminated through posters which were distributed in general practitioners’ offices. Finally, in order to gain access to a larger number of people in precarious situations who do not have Internet access, the questionnaire was distributed in paper form to the social services network of the city of Brest.

Participants who were included were those who completed the questionnaires between 18 January and 9 March 2021. There were no inclusion or non-inclusion criteria: this was a general population observational study relying on the willingness of participants to respond.

### 2.2. Description of Variables and Measures

#### 2.2.1. Sociodemographic Variables

The sociodemographic variables collected were gender, age, place of residence, living conditions (couples with or without children and single parents with children were considered to be families), and socio-professional category. A specific distinction was made between respondents working as health professionals and others. We also collected recent socio-professional status changes during the pandemic and lockdowns.

#### 2.2.2. Medical and Psychiatric History

We explored the history of health disorders: respondents were asked if they had a history of medical or surgical disorders or chronic illness, or psychiatric or addictive disorders. They were also asked if more particularly during the pandemic they had experienced medical/surgical or psychiatric disorders and if they had been affected by COVID-19 in any way.

#### 2.2.3. Main Outcome: Changes in Alcohol Consumption

We explored reported alcohol consumption before, during, and after lockdowns. The questions were all worded to focus on any evolution in consumption with the pandemic and lockdowns.

The AUDIT-C questionnaire was used to assess alcohol consumption. In this questionnaire, three variables are assessed: “frequency of alcohol consumption,” “number of drinks per day,” and “frequency of heavy drinking”(6 or more drinks on a single occasion in the past year). Responses to each item are rated from 0 to 4. The AUDIT-C score was calculated for each participant by summing the responses to the three variables listed above. The score ranges from 0 to 12 (a score of 0 reflects no alcohol use). In men, a score of 4 or more was considered positive; in women, a score of 3 or more was considered positive. The AUDIT-C score was then categorized into three levels: “low-risk use” for a score between 0 and 2 for women and 0 and 3 for men; “probable misuse” for a score above 3 for women and above 4 for men; “probable dependence” for a score above 9 in both genders.

In order to evaluate the participants’ perceptions of their relationship with alcohol, two questions were added: the first one explored the impression of consuming too much alcohol and the second one the impression of having difficulty controlling one’s alcohol consumption. These two items indirectly reflect the notion of loss of control over alcohol consumption.

Finally, as the objective of this study was to evaluate the evolution of alcohol consumption in the context of the pandemic and the lockdowns, a question was asked about the evolution of alcohol consumption in this context: “Do you have the impression that your alcohol consumption evolved with the lockdowns? If so, how?” with four possible choices: increased/decreased/discontinued/no change.

#### 2.2.4. Other Addictive Practices

We explored reported consumption before, during, and after lockdowns. The questions were all worded to focus on the evolution of consumption with the pandemic and lockdowns. 

The use of tobacco, cannabis, and gambling were assessed via the questions: “Do you smoke? Do you, even occasionally, use illicit substances? If so, which one(s)?” with five options: “cannabis/cocaine/heroin/methamphetamines/other,” and “Do you play money and gambling games (scratch cards, raffles, bets, poker)?”.

### 2.3. Statistical Analyses

The data were collected and anonymized in an Excel file. Statistical analyses were performed using R software, version 3.5.3. Type 1 error was set at 5%.

Descriptive statistics were generated to describe the socio-demographic data of the sample and the relevant medical information, and in particular to estimate the prevalence of alcohol use among respondents and changes in consumption in the past year (lockdowns and pandemic).

Univariate and multivariate logistic analyses were conducted to determine which variables were associated with increased, decreased, or discontinued alcohol use. Univariate analyses for categorical variables were performed using Fisher’s exact test. Univariate analyses for quantitative variables were performed using one-way ANOVA or a Kruskal–Wallis test (depending on data distribution). Only variables that were significantly associated with a change in alcohol consumption in the univariate analyses were included in the multivariate logistic analysis. A stepwise backward elimination algorithm was used to eliminate variables that provided little information in the multivariate logistic model.

## 3. Results

### 3.1. Descriptive Statistics

In total, 2491 subjects were included in the study (1123 health professionals, and 1368 non-health professionals from the general population). The statistics for the descriptive analyses for socio-demographic variables, medical and psychiatric history, and other addictive practices are presented in Table 1.

The overall response rate in the general population could not be measured, as the study was circulated online, and through social media and posters in primary care settings. Concerning health professionals specifically, at the time of the study, at the University Hospital of Brest (January to March 2021), where the study was circulated online, 4200 individuals were health professionals, and 3688 (87.5%) were women. Among these health professionals, 1123 responded, giving a response rate of 26.73% for health professionals.

### 3.2. AUDIT-C Score

After calculation of the AUDIT-C score, 48.35% (*n* = 1088) of the sample was classified as having “low-risk use,” 51.65% (*n* = 1090) as having “probable misuse,” and 0.88% (*n* = 19) as having “probable dependence.” If we standardize these estimates to remove skewness towards female gender, we obtain 45.76% of the sample classified as having “low-risk use,” 53.03% as having “probable misuse,” and 1.21% as having “probable dependence”.

Of our sample, 37.61% were female health professionals, 7.41% male health professionals, 37.2% female non-health professionals, and 17.78% male non-health professionals. The median AUDIT-C scores were twice as high for males compared to females regardless of whether or not they were health professionals.

### 3.3. Perceptions of the Relationship with Alcohol

In our study, 31.52% (*n* = 572) of the participants perceived that they drank too much alcohol, and 15.69% (*n* = 285) perceived that they had difficulty controlling their alcohol consumption.

### 3.4. Evolution of Alcohol Consumption

Of the subjects interviewed, 27.64% (*n*= 583) reported having increased their alcohol consumption, it decreased for 14.7% (*n* = 310) and was discontinued for 3.94% (*n* = 83), and 53.72% reported that they had not modified their alcohol consumption.

### 3.5. Variables Associated with an Increase in Alcohol Consumption Compared to Stable Consumption

A preliminary analyses selected independent variables before reaching the final model.

The results of the multivariate logistic analysis for factors associated with increased alcohol consumption compared to stable consumption are presented in Table 2.

The likelihood to have increased alcohol use was associated with being a man (OR = 1.47 (1.14–1.90), *p* < 0.001), being between 45 and 65 years of age (OR = 2.9 (1.45–6.36), *p* < 0.001), having a family (OR = 0.62 (0.46–0.83), *p* < 0. 001, for those living alone versus families), not being a health professional (OR = 0.79 (0.64–0.99), *p* = 0.039), having had a health problem during lockdown (OR = 1.66 (1.32–2.08), *p* < 0.001), smoking tobacco (OR = 1.54 (1.21–1.96), *p* < 0.001), and using cannabis (OR = 3.04 (1.93- 4.87), *p* < 0.001).

### 3.6. Variables Associated with Reduced or Discontinued Alcohol Consumption Compared to Stable Consumption

A preliminary analyses selected independent variables before reaching the final model.

They detail how independent variables were selected before reaching the final model.

Since the number of respondents who ceased alcohol consumption was small, they were combined with respondents who reduced their alcohol consumption. The results of the multivariate logistic analysis for factors associated with decreased or discontinued alcohol consumption compared to stable consumption are presented in Table 3.

The likelihood to have reduced or discontinued drinking was associated with being a man (OR = 1.57 (1.18–2.09), *p* < 0.001), not living in the city (Brest), (OR = 1.52 (1.18–1.96), living alone (OR = 1.58 (1.16–2.15), <0.001, for those living alone versus families), and being student and using cannabis (OR = 2.93 (1.79–4.82), *p* < 0.001).

## 4. Discussion

In this general population study, we obtained broad participation in the general population with 2491 responses, and more particularly among health professionals, with 1123 participants. Our population was mainly composed of women and working individuals. Alcohol consumption and alcohol misuse were markedly prevalent. Indeed, 51.65% of our sample presented probable misuse of alcohol. This result is nearly twice as high as alcohol misuse prevalence in the general population in France [8,19]. In addition, 31.52% of the participants in our study reported that they had the feeling that they were drinking too much alcohol, and 15.69% had the impression of having difficulty controlling their alcohol consumption. Concerning the evolution of alcohol consumption with the pandemic and lockdowns, half of the sample reported no change (53.7%) but among those who reported a change, the majority (27.6% of the whole sample) had increased their alcohol consumption, while 14.7% had decreased and 3.94% had stopped.

Our results on alcohol consumption contrast with those of Kilian et al. [17], who in their meta-analysis showed a trend towards a decrease in alcohol consumption during the first months of the COVID-19 pandemic in Europe, and with the results obtained by Guignard et al. in France [18]. In a study conducted in Israel [20], Bonny-Noach et al. showed a positive association between the duration of the pandemic and the number of lockdowns and increased alcohol consumption. We can hypothesize that alcohol consumption rebounded after the lockdowns because of a gradual return to the usual modes and places of consumption, in addition to the alcohol consumption as a coping strategy to manage psychological distress. It is also possible that the successive periods of lockdown opened the way to new modes of purchase and consumption and, as an example, sales in the United States increased by 234% at the beginning of the pandemic [21,22].

In contrast, only 0.88% met the criteria for probable dependence, 3 times lower than expected if we refer to the estimated prevalence of 3.3% of alcohol use disorders in the French population [1]. This could be explained by our population profile: a majority of women working. This overrepresentation of women is explained by two factors. Women are more concerned about their health and tend to more readily participate in this type of study. A one-month alcohol abstinence campaign in Europe also mobilized a majority of women and working people [23]. Secondly, the questionnaire was widely circulated to a population of health professionals, and in 2018, 90% of public hospital employees in France were women [24], and in the University Hospital of Brest at the time of the study 87.5% of the health professionals were women. Another hypothesis concerns an underestimation by respondents of their consumption because of stigmatization, which is higher in women for substance use [25,26]. In addition, the low level of probable alcohol dependence could also be related to the selected tool: AUDIT-C is a screening test but not a diagnostic test [26,27,28].

We observed that people aged 45 to 65 and living with their family increased their alcohol consumption more than others. In contrast, being a student (compared to employees, retired, or inactive) and being urban (living in the city of Brest) were associated with cessation or reduction in alcohol use. Decreased alcohol use among young people and students has been previously reported during lockdowns [29,30]. One hypothesis could be linked to changes in alcohol consumption habits. Reduced access to usual social drinking venues such as bars, restaurants, and night-clubs led to decreased drinking among young people and urban residents. Another hypothesis could be linked to financial restrictions, since people under 25 and students had greater financial difficulties during lockdown, as a result of restricted access to jobs and isolation from the family. This could explain reduction or cessation of alcohol consumption.

People living alone were more likely to have decreased their alcohol consumption compared to those living with family. Social isolation is usually a risk factor for alcohol misuse, but Killgore et al. also observed that the greatest increase in high-risk drinking over the course of the pandemic occurred predominantly among individuals confined at home close to family members [29]. During the lockdowns, schools and nurseries were closed and parents had to ensure that children stayed involved in educational activities, while at the same time managing work-from-home responsibilities and general housework. Having children at home during the pandemic increased psychological distress [30]. Working at home was also a fundamental change. Workers who would never have thought of consuming alcohol at the office were now free to drink to excess during work hours. These alcohol use behaviors could be responses to cope with a multifactorial, anxiety-ridden context arising from the stresses of job loss and other economic hardships, social isolation, disrupted routines, and the general uncertainties and anxieties concerning the virus itself. This type of evolution could lead, in the medium and long term, to reduced motivation for work, increased errors and poor judgment, or absenteeism.

In addition, the isolation and overcrowding caused by the lockdowns and working from home could have led to relationship problems, all the more so if combined with alcohol consumption [31]. We did not explore domestic violence in this study, but Piquero et al. [32] showed a sharp increase in domestic violence and Rodriguez et al. also showed an increased risk of child abuse between before and after lockdowns [33].

Having had a medical or psychiatric disorder during the lockdowns was also associated with increased alcohol consumption, which seems consistent with previous studies. This increase could reflect alcohol consumption for anxiolytic purposes or to combat depressive affects [7,8].

Finally, men and individuals with current consumption of tobacco and/or cannabis tended to have increased alcohol consumption in our sample results. For these people more at risk of developing alcohol use disorder, Pabst et al. reported the greatest variations in absolute values of quantities consumed before and after lockdowns [34]. People with a substance use disorder reduced or increased their alcohol consumption in units per week, respectively, five and eight times more than others.

The present study enables a focus on health professionals. First, the response rate was high, as was the spontaneous participation of health professionals. We disseminated the questionnaire through the mailing list of the university hospital, and the response rate was close to a quarter of the population of professionals. This reflects a real concern among health professionals for their health and addictive disorders. In our study, they appeared less likely to increase their alcohol consumption, but other studies have reported an increase in alcohol consumption for more than 50% of medical staff during the first lockdown [35].

Health professionals were exposed to specific risk factors. They were directly exposed to stress factors resulting from the virus and the lockdowns, and they had to cope with a complete, rapid reorganization of the French healthcare system. Consumption patterns could reflect an adaptation of healthcare professionals in the course of the pandemic, as they learned to cope and work with COVID-19. However, the composition of our sample and the diversity of the health professionals surveyed means that caution is required, and more specific studies are necessary.

### Strengths and Limitations

One strength of our study is the sample size and diversity of the variables analyzed, as well as the temporality of data collection. To our knowledge, few studies have examined the longer-term impact of the pandemic on alcohol consumption.

Nevertheless, the limitations of the study should also be considered. The cross-sectional nature of our study is a limiting factor. The analyses in the study were based on data from self-administered online questionnaires, and questions about past events may have led to a recall bias. The questionnaire was completed by a large sample of health professionals so that the composition of our sample was not representative of the general population. Indeed, the sample mainly consisted of women, and working individuals. Finally, the sample was recruited in a single region of France (Britanny), cultural bias may exist, and our results could not be generalized to the whole French population.

## 5. Conclusions

In this study, we collected data on alcohol consumption in the context of a pandemic and lockdowns in a large sample in a French region. The majority of the sample was constituted of working women. More than half of our population responded as having probable alcohol misuse, and a majority did not change alcohol consumption since the lockdowns. However, among those who declared a change, the majority increased. Different variables were associated with the likelihood to have increased alcohol consumption, such as being a man, living with family, being middle aged, or having medical or addictive history. On the contrary, urban residents and students were more likely to reduce their alcohol consumption. However, these results could not be generalized, and further studies are required to confirm these constatations.

## Figures and Tables

**Table 1 ijerph-19-15049-t001:** Descriptive statistics for socio-demographic variables, medical and psychiatric history, and other addictive practices.

	Alcohol Consumption				
	Increased	No Change	Decreased	Ceased	Group Comparison *
	*n* (%)	*n* (%)	*n* (%)	*n* (%)	*p*-Value
Gender					
Male	166 (28.67)	241 (41.63)	104 (17.96)	68 (11.74)	<0.001
Female	417 (27.25)	892 (58.30)	206 (13.46)	15 (0.98)	
Age group					
18–25	41 (15.83)	114 (44.01)	81 (31.27)	23 (8.88)	<0.001
26–44	346 (33.3)	489 (47.10)	161 (15.51)	42 (4.04)	
45–65	186 (24.50)	490 64.55)	66 (8.73)	17 (2.23)	
>65	10 (18.86)	40 (75.47)	2 (3.77)	1 (1.88)	
Place of residence in Brest					
No	342 (28.12)	691 (56.82)	145 (11.92)	38 (3.12)	<0.001
Yes	241 (26.98)	442 (49.49)	165 (18.47)	45 (5.03)	
Living conditions					
Family	468 (30.19)	849 (54.77)	182 (11.74)	51 (3.29)	<0.001
Flat-sharing	21 (15.55)	65 (48.14)	42 (31.11)	7 (5.18)	
Alone	89 (22.53)	203 (51.39)	79 (20.00)	24 (6.07)	
Socio-professional category					
Student	21 (14.58)	65 (45.13)	44 (30.55)	14 (9.72)	<0.001
Employee	459 (29.14)	844 (53.58)	220 (13.96)	52 (3.30)	
Self-employed	34 (36.95)	38 (41.30)	16 (17.39)	4 (4.34)	
Not working	22 (26.50)	48 (57.83)	8 (9.63)	5 (6.02)	
Retired	18 (17.64)	75 (73.52)	4 (3.92)	5 (4.90)	
Health professionals					
Yes	235 (24.65)	540 (56.66)	139 (14.58)	39 (4.09)	0.039
No	348 (30.10)	593 (51.29)	171 (14.79)	44 (3.80)	
Changes in socio-professional status					
Studying from home	24 (18.32)	58 (44.27)	38 (29.00)	11 (8.39)	<0.001
Working from home	124 (31.15)	198 (49.74)	62 (15.57)	14 (3.51)	
Temporary unemployment	34 (36.17)	41 (43.61)	18 (19.14)	1 (1.06)	
Job loss	16 (34.78)	18 (39.13)	9 (19.56)	3 (6.52)	
No change	34 (21.11)	88 (54.65)	34 (21.11)	5 (3.10)	
Increased workload	177 (34.57)	246 (48.04)	67 (13.08)	22 (4.29)	
Somatic comorbidity					
Yes	173 (27.15)	365 (58.86)	67 (10.51)	32 (5.02)	0.002
No	410 (27.85)	768 (52.17)	243 (16.50)	51 (3.46)	
Psychiatric or addictive comorbidity					
Yes	51 (36.69)	71 (51.07)	11 (7.91)	6 (4.31)	0.024
No	532 (27.00)	1062 (53.90)	299 (15.17)	77 (3.90)	
Health problem during lockdowns					
Yes	228 (33.04)	322 (48.11)	106 (15.36)	34 (4.92)	<0.001
No	355 (25.01)	811 (57.15)	204 (14.37)	49 (3.45)	
Exposure to COVID-19					
Personal harm	13 (22.80)	32 (56.14)	10 (17.54)	2 (3.50)	<0.001
Harm to a relative	162 (28.42)	244 (42.80)	93 (16.31)	21 (3.68)	
Death of a relative	15 (28.30)	25 (47.16)	10 (18.86)	3 (5.66)	
Not concerned	376 (25.64)	807 (56.39)	192 (13.41)	56 (3.91)	
Smoking tobacco					
Yes	214 (36.70)	254 (43.56)	97 (16.63)	18 (1.25)	<0.001
No	369 (24.22)	877 (57.58)	212 (13.91)	65 (4.26)	
Using cannabis					
Yes	75 (48.70)	33 (21.42)	42 (27.27)	4 (2.59)	<0.001
No	508 (25.98)	1100 (56.26)	268 (13.70)	79 (4.04)	
Gambling					
Yes	156 (28.94)	285 (52.87)	76 (14.10)	22 (4.08)	0.866
No	426 (27.26)	848 (54.01)	233 (14.84)	61 (3.88)	

* Univariate group comparisons were performed using Fisher’s exact test.

**Table 2 ijerph-19-15049-t002:** Multivariate logistic analysis: crude and adjusted odds ratios (OR_c_, OR_a_) and 95% confidence interval (CI) for individuals with increased alcohol consumption compared to those with stable alcohol consumption.

	OR	95% CI−	95% CI+	*p*-Value
Gender				
Female	1.00		Ref.	
Male	1.47	1.14	1.9	<0.001
Age				
18–25	1.00		Ref.	
26–44	1.59	0.713	3.78.	0.273
45–65	2.9	1.45	6.36	<0.001
>65	1.58	0.713	3.46	0.224
Living conditions				
Family	1.00		Ref.	
Flat-sharing	0.647	0.35	1.16	0.153
Alone	0.617	0.455	0.831	<0.001
Caregiver				
No	1.00		Ref.	
Yes	0.794	0.638	0.988	0.038
Health problem during lockdowns				
No	1.00		Ref.	
Yes	1.66	1.32	2.08	<0.001
Smoking tobacco				
No	1.00		Ref.	
Yes	1.54	1.21	1.96	<0.001
Using cannabis				
No	1.00		Ref.	
Yes	3.04	1.93	4.87	<0.001

**Table 3 ijerph-19-15049-t003:** Multivariate logistic analysis: adjusted ORs and 95% CIs for individuals who reduced or ceased alcohol consumption compared to stable alcohol consumption.

	OR	95% CI−	95% CI+	*p*-Value
Gender				
Female	1.00		Ref.	
Male	1.57	1.18	2.09	<0.001
Place of residence				
Brest	1.00		Ref.	
Other	1.52	1.18	1.96	<0.001
Living conditions				
Family	1.00		Ref.	
Flat-sharing	1.69	1.05	2.69	0.029
Alone	1.58	1.16	2.15	<0.001
Socio-professional category				
Student	1.00		Ref.	
Employee	0.562	0.363	0.872	<0.001
Self-employed	0.89	0.439	1.77	0.743
Not working	0.26	0.117	0.544	<0.001
Retired	0.179	0.075	0.387	<0.001
Using cannabis				
No	1.00		Ref.	
Yes	3.26	1.96	5.45	<0.001

## Data Availability

All data generated or analyzed during this study are included in this published article.

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
