# Peer review of "Changes in Alcohol Consumption after 1 Year of the COVID-19 Pandemic: A Cross-Sectional Study in a Region of France"

_ijerph, 2022, doi:10.3390/ijerph192215049_

Round 1

Reviewer 1 Report (Previous Reviewer 1)

Dear authors,

Thank you for taking my comments into account and adapting your manuscript, accordingly. I have no further suggestions apart from publishing your works.

Author Response

many thanks for your comment

This manuscript is a resubmission of an earlier submission. The following is a list of the peer review reports and author responses from that submission.

Round 1

Reviewer 1 Report

  • Summary

    • Villette and colleagues examine a very important cultural aspect for the health sector in respect to the COVID19 pandemic. Alcohol abuse is one of the many coping mechanisms in such stressful situations and should be monitored frequently and sensitively. In that respect, this paper is a much-needed and timely perspective.
  • Major comments

    • Overview of the return percentages according to the different lines of distribution
    • The distribution between sexes is strongly skewed towards females. Please comment on this and discuss your results in the light of this factor. In addition to your results thus far, please normalize your results by the population distribution in order to correct for this influence. Women are more self-reflecting, alcohol abuse is more stigmatized in women - this might be a simple explanation for your increase in misuse in the self-reporting AUDIT-C as well as the low numbers of dependence. Arguments as in line 302 and following hinge upon this fact and should be reframed after analysis. Merely explaining the confounding effects (line 308,325) does not help with the actual interpretation.
    • It seems that although multivariate analysis was employed, no interaction between factors was looked upon. Please provide the full analysis as supplement, also include interaction terms into the analysis in order to compare influences (e.g.using multilinear modeling). Sentences like line 274 and 290 are baseless without these interaction terms. Here also the skewed female to male percentages might play into the results. Re-run the analysis using the correction term calculated earlier.
    • PLease display the percentages of caretaking women, other women, caretaking men and other men in a simple bar graph. As you discussed, this is vital information to understand the collective and outcomes. You might accompany this display with the Audit-C scores for the respective groups and/or calculated estimated marginal means for Audit-C scores in these subgroups.
  • Minor comments

    • Please describe what multivariate analysis was used for the comparison of the significantly different items. Multivariate as such is not enough to be able to reproduce these results.
    • It might be interesting how the percentage of alcohol- drinking participants relates to the estimated alcohol- dependent population. Or is this number derived from this very questionnaire?
    • Please correct the grammar and spelling in the document (e.g. line 225 "twice-fold" should be "two-fold", sentence in line 250 (missing words)).

Author Response

Responses to reviewers

Villette and colleagues examine a very important cultural aspect for the health sector in respect to the COVID19 pandemic. Alcohol abuse is one of the many coping mechanisms in such stressful situations and should be monitored frequently and sensitively. In that respect, this paper is a much-needed and timely perspective.

R : Many thanks for this comment

Reviewer 1

Major comments

  • Overview of the return percentages according to the different lines of distribution
  • The distribution between sexes is strongly skewed towards females. Please comment on this and discuss your results in the light of this factor. In addition to your results thus far, please normalize your results by the population distribution in order to correct for this influence. Women are more self-reflecting, alcohol abuse is more stigmatized in women - this might be a simple explanation for your increase in misuse in the self-reporting AUDIT-C as well as the low numbers of dependence.
  • R: our results are really interesting face to this comment, we collected informations regarding alcohol consumption in a population, that usually is not identified as having alcohol misuse and that doesn't seek help
  •  Arguments as in line 302 and following hinge upon this fact and should be reframed after analysis. Merely explaining the confounding effects (line 308,325) does not help with the actual interpretation.
  • R : we tried to develop this point population in the discussion. But regarding the French proportion of women in health givers, which can achieve 90% of the population, and the fact that we diffused the questionnaire in a large sample of health givers the proportion of women is not surprising. "Two factors could explain this representation : the first is that women are more concerned with their health (REF)  , and tend to more easily participate to this type of study through self-questionnaire . Studies in Europe on one-month alcohol abstinence campaign, which enhance the general population to reflect and change their alcohol consumption , mobilized a majority of women, who worked , comparable to our ( De ternay J Harm Red journal 2022 )  The second is that the questionnaire was widely communicated in a population of health professionals, and in 2018, 9 /10 of employees of public hospital were women (1)"Although more men did not participate to the study, the number of responses is still large enough to make reliable estimations. We added both crude and adjusted OR estimates. Adjustments are made using sex and age class as well as other independent variables that showed a sufficient explanatory power (defined as a p-value >0.20). This is detailed in Supplementary materials.
  • It seems that although multivariate analysis was employed, no interaction between factors was looked upon. Please provide the full analysis as supplement, also include interaction terms into the analysis in order to compare influences (e.g.using multilinear modeling). Sentences like line 274 and 290 are baseless without these interaction terms. Here also the skewed female to male percentages might play into the results. Re-run the analysis using the correction term calculated earlier.

R: R : we understand that the statistical concerns. However, hypothesis regarding changes in alcohol consumption habits are valuable , and the fact that changes in alcohol consumption habits can be maintained in time are consistent too. It may of course not explain the whole but these hypothesis need to be developed according to us. We eliminated independent variables that had a strong link between each other if after being adjusted to each other they lose their explanatory (please refer to the answer to the previous question).

  • PLease display the percentages of caretaking women, other women, caretaking men and other men in a simple bar graph. As you discussed, this is vital information to understand the collective and outcomes. You might accompany this display with the Audit-C scores for the respective groups and/or calculated estimated marginal means for Audit-C scores in these subgroups.

R: we added two figures to address these interrogations.

Minor comments

  • Please describe what multivariate analysis was used for the comparison of the significantly different items. Multivariate as such is not enough to be able to reproduce these results.

R: we added the word logistic. Besides, following first reviewer’s comments, we also provided more details on preliminary steps that led to the retained models (supplementary material).

  • It might be interesting how the percentage of alcohol- drinking participants relates to the estimated alcohol- dependent population. Or is this number derived from this very questionnaire?
  • Please correct the grammar and spelling in the document (e.g. line 225 "twice-fold" should be "two-fold", sentence in line 250 (missing words)).

Reviewer 2 Report

In the current article authors investigated the alcohol consumption one year after the onset of the COVID-19 pandemic in France. The article deals with an important aspect and was well-written. I have some comments.

First of all, authors conclude that the COVID-19 pandemic seems to have led to a long-term increase in alcohol consumption. However, I do not feel that the results support such a statement. According to study results 27.64% of participants increased alcohol consumption, 14.7% decreased, 3.94% ceased and 53.72% did not modify their alcohol consumption. In other words, about ¼ of the population reported increased alcohol consumption.

Moreover, authors should explain more in Discussion the main findings of the study, ie why some factors were more associated either with increased or decreased alcohol consumption.

Cannabis use was positively associated with lower and higher consumption of alcohol. This is a non-expected finding. Authors should explain more this finding in Discussion. Additionally, it would be interesting to check if this association remains the same after splitting their sample to younger and older adults or to men and women.

There were many caregivers that participated in the study. Is it possible that the questionnaire was disseminated through sites regarding aspects like health and care of older people?

In section 2.2.3., lines 141-142 there is a typo. Authors report that “people with a score higher that 12 were considered in a state of probable dependence”. However, the max score of AUDIT-C is 12.

Author Response

Reviewer 2

  • In the current article authors investigated the alcohol consumption one year after the onset of the COVID-19 pandemic in France. The article deals with an important aspect and was well-written. I have some comments.
  • R : many thanks for this comment
  • First of all, authors conclude that the COVID-19 pandemic seems to have led to a long-term increase in alcohol consumption. However, I do not feel that the results support such a statement. According to study results 27.64% of participants increased alcohol consumption, 14.7% decreased, 3.94% ceased and 53.72% did not modify their alcohol consumption. In other words, about ¼ of the population reported increased alcohol consumption.
  • R : we changed this conclusion, reviewer is right, a majority did not change alcohol consumption, but among those who changed, the tendency was towards an an increase of alcohol consumption.
  • Moreover, authors should explain more in Discussion the main findings of the study, ie why some factors were more associated either with increased or decreased alcohol consumption.
  • R : we tried to insist on the main findings in the discussion, we changed the discussion
  • Cannabis use was positively associated with lower and higher consumption of alcohol. This is a non-expected finding. Authors should explain more this finding in Discussion. Additionally, it would be interesting to check if this association remains the same after splitting their sample to younger and older adults or to men and women.

R: The reference group is always the group that reports a stable consumption.

  • Age class and sex were included in our final model on increase vs stable alcohol consumption.
  • Age class and sex were included in our final model on reduced/ceased vs stable alcohol consumption.

The impact of age class and gender on OR can be evaluated by comparing crude and adjusted OR.

Cannabis use was associated with changes in alcohol consumption.

  • There were many caregivers that participated in the study. Is it possible that the questionnaire was disseminated through sites regarding aspects like health and care of older people?
  • R : In the method section, we explain how the questionnaire was disseminated. The study was disseminated through the press (communication in the local press and on television) and on the Internet (social networks and mailing lists of the investigators, and also the city of Brest, the University Hospital of Brest, the University of Western Brittany and in particular the Citizen's University of Health Prevention of Western Brittany, the North Finistere Addictology Network and the Department of General Medicine). It was disseminated through posters: posters were distributed in general practitioners' offices. Finally, in order to gain access to a larger number of people in precarious situations who do not have Internet access, the questionnaire was distributed in paper form to the structures in the social services network of the city of Brest. The response rate in the university hospital of Brest was high, we added elements concerning this healthcare population in the result section : Focusing on health professionals, at the moment of the study, at the University Hospital of Brest ( January to march 2021),where the study was distributed online, 4200 persons were health professionals , and 3688 (87.5%) were women. Among the health professionals, 1123 persons had responded , so the response rate was 26.73% among health professionals.
  • In section 2.2.3., lines 141-142 there is a typo. Authors report that “people with a score higher that 12 were considered in a state of probable dependence”. However, the max score of AUDIT-C is 12.
  • R : reviewer is right, Audit c score above 10 corresponded to a higher risk of dependence . we changed it.

Reviewer 3 Report

The study objective is important and novel.

However, some changes are needed:

1. The manuscript requires moderate language changes.

2. The methods section - please clarify the way of distribution of the questionnaire. How the sample was selected, inclusion/exclusion criteria, etc.

3. The following section is unclear and requires changes to provide more informative data "2.1.1. Health-related variable"

4. If available, please provide information on the response rate

5. There is no need to divide results into short sub-sections that provides only 2 sentences (e.g. 3.2;3.3.;3.4)

6. Please add 2-3 sentences on practical implications and further research needs

Author Response

  • Reviewer  3
  • the study objective is important and novel.
  • R : many thanks
  • However, some changes are needed:
  • The manuscript requires moderate language changes.
  • R : the manuscript was reviewed by an english native before submission, but this revised version was once more reviewed by an english native,as recommended by reviewers and guest editor

  • The methods section - please clarify the way of distribution of the questionnaire. How the sample was selected, inclusion/exclusion criteria, etc.
  • Distribution of the questionnaire
  • The inclusion criteria were : over 18 years old, French language comprehension ( oral and written), and consent to participate. We wanted to target the adult general population, so we did not exclude participants, except those under 18.
  • The questionnaire was widely disseminated to the general population through several media, either through e mails list with a a link to the study, or through a QR code which was linked to the questionnaire. We collected the agreement of the direction of the university hospital of Brest, the direction of the city of Brest and of the University to disseminate the questionnaire to employees through the mailing list. The study was disseminated through the press (communication in the local press and on local television) and on the Internet (social networks and mailing lists of the investigators, and also the city of Brest, the University Hospital of Brest, the University of Western Brittany and in particular the Citizen's University of Health Prevention of Western Brittany, the North Finistère Addictology Network and the Department of General Medicine). It was disseminated through posters: posters were distributed in general practitioners' offices. Finally, in order to gain access to a larger number of people in precarious situations who do not have Internet access, the questionnaire was distributed in paper form to the structures in the social services network of the city of Brest.

  • The following section is unclear and requires changes to provide more informative data "2.1.1. Health-related variable"
  • R : we completed
  • If available, please provide information on the response rate
  • R : we completed the results section, we can measure the response rate for health care providers, but not for the general population
  • There is no need to divide results into short sub-sections that provides only 2 sentences (e.g. 3.2;3.3.;3.4)
  • We made changes
  • 6. Please add 2-3 sentences on practical implications and further research needs
  • R we tried to develop this point
